# Machine Learning-Based Prediction of Heatwave-Related Hospitalizations: A Case Study in Matam, Senegal

**DOI:** 10.3390/ijerph22091349

**Published:** 2025-06-23

**Authors:** Mory Toure, Ibrahima Sy, Ibrahima Diouf, Ousmane Gueye, Endalkachew Bekele, Md Abul Ehsan Bhuiyan, Marie Jeanne Sambou, Papa Ngor Ndiaye, Wassila Mamadou Thiaw, Daouda Badiane, Aida Diongue-Niang, Amadou Thierno Gaye, Ousmane Ndiaye, Adama Faye

**Affiliations:** 1Agence Nationale de l’Aviation Civile et de la Météorologie (ANACIM), Dakar BP 8184, Senegal; papangor.ndiaye@anacim.sn (P.N.N.); aida.diongue.niang@anacim.sn (A.D.-N.); 2Laboratoire de Physique de l’Atmosphère et de l’Ocean–Simeon Fongang (LPAO-SF), Ecole Superieure Polytechnique, Universite Cheikh Anta Diop (UCAD), Dakar BP 5085, Senegal; ibrahima.diouf@univ-labe.edu.gn (I.D.); marie.sambou@ucad.edu.sn (M.J.S.); daouda.badiane@esp.sn (D.B.); atgaye@esp.sn (A.T.G.); 3Ministère de la Santé et de l’Action Sociale, Dakar BP 4024, Senegal; ibrahima.ousmane.sy@gmail.com; 4Departement de Geographie, Universite Cheikh Anta Diop (UCAD), Dakar BP 5005, Senegal; 5Centre de Suivi Ecologique, Dakar BP 15532, Senegal; 6Faculté des Sciences et Techniques, Université de Labe, Labe BP 210, Guinea; 7Centre Hospitalier Régional El Hadji Ibrahima Niass (CHREIN), Kaolack BP 24030, Senegal; gueye_99@yahoo.fr; 8National Center for Environmental Prediction (NCEP), National Oceanic and Atmospheric Administration (NOAA), College Park, MD 20740, USA; endalkachew.bekele@noaa.gov (E.B.); ehsan.bhuiyan@noaa.gov (M.A.E.B.); wassila.thiaw@noaa.gov (W.M.T.); 9African Center of Meteorological Applications for Development (ACMAD), Niamey BP 13184, Niger; ousmane.ndiaye@acmad.org; 10Institut de Santé et Développement (ISED), Université Cheikh Anta Diop (UCAD), Dakar BP 5005, Senegal; adamafaye94@gmail.com

**Keywords:** heatwave, hospital admissions, machine learning, climate-health, early warning systems, Senegal

## Abstract

This study assesses the impact of heatwaves on hospital admissions in the Matam region of Senegal by combining climatic indices with machine learning methods. Using daily maximum temperature (TMAX) and heat index (HI), heatwave events were identified from 2017 to 2022. Hospital data from Ourossogui Regional Hospital were analyzed, and three predictive models, Random Forest (RF), Extreme Gradient Boosting (XGB), and Generalized Additive Models (GAMs), were compared. A bootstrapping approach with 1000 iterations was used to evaluate model robustness. The findings reveal a significant delayed effect of heatwaves, with increased hospitalizations occurring three to five days after the event. RF outperformed the other models with R^2^ values ranging from 0.51 to 0.72. These findings highlight the need to enhance heatwave monitoring and promote the integration of impact-based climate forecasting into health early warning systems, particularly to protect vulnerable groups such as the elderly, children, and outdoor workers.

## 1. Introduction

Climate change is now an undeniable reality, with major impacts on ecosystems, societies, and, more particularly, human health [1,2]. According to the Sixth Assessment Report of the Intergovernmental Panel on Climate Change [3], the frequency, duration, and intensity of extreme temperatures have significantly increased worldwide, exacerbating health risks, especially for vulnerable populations. The World Health Organization [4] estimates that heat stress could become one of the leading causes of climate-related mortality in the coming decades, particularly in sub-Saharan Africa, where access to healthcare remains limited [5,6].

Although Africa contributes minimally to global greenhouse gas emissions, it disproportionately suffers from the effects of climate change, particularly in Sahelian regions [7,8]. The Sahel is identified as one of the most climate-sensitive areas [9], with projected increases in average temperatures ranging from 1.5 °C to 6.5 °C by the end of the 21st century, depending on different greenhouse gas emission scenarios [10]. This temperature rise is accompanied by an intensification of extreme weather events, particularly heatwaves and droughts [10,11], exposing populations to severe health consequences.

Heatwaves have become a major public health issue, particularly in arid and semi-arid regions such as the Sahel [12]. They lead to a significant increase in morbidity and mortality, especially among vulnerable populations [13,14,15]. These climate events can cause or exacerbate certain pathologies, including heat stroke, dehydration, cardiovascular diseases, and respiratory illnesses [16,17].

Beyond their direct effects, heatwaves place considerable pressure on healthcare infrastructure, especially in areas with limited access to medical care [18,19]. They also compromise food security and access to clean water, further increasing the prevalence of infectious and nutritional diseases [20,21].

In response to these challenges, several initiatives have been implemented to improve heatwave management and mitigate their health impacts. In Senegal, the Agence Nationale de l’Aviation Civile et de la Météorologie (ANACIM), in collaboration with the National Oceanic and Atmospheric Administration (NOAA), the Senegalese Ministry of Health, the Senegalese Red Cross, and the Centre de Suivi Ecologique (CSE), has been working since 2016 to establish monitoring and early warning systems for heatwaves and their effects on public health [22]. However, these efforts remain limited, particularly in establishing precise links between heatwaves and increased hospitalizations. The lack of consolidated health data hinders the formulation of appropriate recommendations and the determination of region-specific alert thresholds.

Globally, the increasing occurrence of heatwaves has led to a sharp rise in hospitalizations and deaths, as demonstrated by the dramatic 2003 heatwave in Europe, which caused thousands of fatalities [23]. This is consistent with findings in multicountry studies showing that both heat and cold significantly impact mortality trends [24,25]. In Africa, these events are becoming more frequent, exacerbating their public health consequences [26].

The impacts of heatwaves vary depending on several socio-demographic factors. Elderly and dependent individuals are particularly vulnerable due to their reduced thermoregulation capacity and the higher prevalence of chronic diseases in this age group [27]. Infants and young children are also at risk due to the immaturity of their thermoregulation system, making them more susceptible to dehydration and febrile seizures [28]. Outdoor workers, including construction workers, farmers, and herders, are exposed to prolonged high temperatures, increasing their risk of heat stress and heat-related illnesses [29]. People with chronic diseases such as cardiovascular, respiratory, or metabolic disorders are more likely to experience health deterioration during extreme heat episodes [30]. Lastly, individuals with reduced mobility, whether due to physical or cognitive disabilities, face greater difficulties adapting to extreme conditions and accessing healthcare [31].

Other environmental factors amplify the effects of heatwaves. High humidity reduces the efficiency of sweating, exacerbating heat stress and increasing the risk of hyperthermia [32]. Additionally, atmospheric dust and fine particulate matter during extreme heat periods can worsen respiratory and cardiovascular diseases, particularly asthma and chronic obstructive pulmonary disease [33].

Given the complexity of interactions between climate and health, it is essential to adopt analytical approaches capable of capturing these multiple dynamics. However, traditional statistical methods often have limitations in accurately assessing these relationships [34], necessitating the exploration of more suitable approaches. Artificial intelligence, particularly machine learning, provides powerful solutions for improving the analysis of climate impacts on health.

The application of machine learning enables the detection and characterization of heatwaves by leveraging algorithms such as the Patient Rule Induction Method (PRIM) and Multivariate Adaptive Regression Splines (MARSs) [35], as well as Random Forest (RF) [36]. These techniques facilitate the identification of crit-ical periods and climate thresholds that may have significant health consequences.

Moreover, these methods are also used to predict hospitalizations linked to heatwaves [37,38]. They allow for the integration of delayed and cumulative effects of extreme temperatures on human health, improving forecasting accuracy and facilitating the implementation of early warning systems [39].

Although several studies have explored the impact of heatwaves on health in West Africa and Senegal, significant gaps remain. For instance, Sambou et al. [40] analyzed heatwaves in Senegal without integrating health data, limiting the assessment of direct links between extreme heat and morbidity. Similarly, Diouf et al. [41] studied the impact of atmospheric dust on these phenomena but did not examine their repercussions on hospitalizations. Additionally, existing research primarily relies on traditional statistical methods, which struggle to capture the complexity of interactions between extreme temperatures, humidity, and health vulnerabilities, posing a major limitation in understanding the real impacts of heatwaves on populations [15].

The objective of this study is to analyze the impact of heatwaves on hospitalizations in Matam from 2017 to 2022. Unlike previous studies that have primarily focused on characterizing heatwaves or assessing their climate trends, this research innovates by exploring the links between extreme heat episodes and hospital admissions, incorporating advanced analytical approaches.

More specifically, this study aims to identify critical thresholds of maximum temperature and optimal combinations of temperature and humidity associated with a significant increase in hospital admissions. It also distinguishes itself by integrating socio-demographic variables (age, gender) to better understand the most vulnerable groups and improve the accuracy of predictive models. The selection of explanatory variables is optimized using advanced selection methods to minimize statistical noise and enhance model robustness.

In terms of methodology, this study evaluates the performance of machine learning compared to traditional statistical approaches in predicting heatwave-related hospitalizations. Three models are tested: a traditional statistical model and two machine learning models. This comparison assesses the extent to which machine learning models outperform conventional approaches in predictive accuracy and in capturing nonlinear interactions between climate and health variables.

The scientific contribution of this research lies in the integration of artificial intelligence techniques to model the complex relationships between climate and health—an approach that remains underexplored in the Sahelian context. By providing relevant and actionable results, this study aims to enhance the existing early warning system and guide public health policies, considering the climatic and socio-economic specificities of the Matam region.

## 2. Materials and Methods

### 2.1. Study Area

Matam, located in northeastern Senegal (see Figure 1), is one of the hottest regions in the country, with extreme temperatures reaching 44–45 °C between March and June [42,43]. Its semi-arid climate, characterized by a dominant dry season and low annual precipitation (369 mm/year) [42,44], exacerbates the frequency of droughts, increasing health and economic risks for the population.

The region has a young and rapidly growing population, with 60% under the age of 25 [45]. Access to healthcare remains limited, with a high maternal mortality rate (252 deaths per 100,000 births) and a high prevalence of cardiovascular, respiratory, and infectious diseases [45]. More than half of the population lacks adequate sanitation, and 82.8% of rural households still rely on firewood for cooking.

### 2.2. Data

The data used in this study come from two main sources:Climate data from the Matam meteorological station, established in 1918.Health data, collected from Ourossogui Regional Hospital, the region’s largest healthcare facility, receives the highest number of patients and the most severe cases.

#### 2.2.1. Climate Data

The climate data used consist of three relevant variables for detecting heatwaves with potential health impacts. These variables were selected primarily based on their availability over time. The selected variables are daily maximum temperature (TMAX) and minimum relative humidity (RHMIN). These parameters were chosen for their relevance in identifying heatwaves and their potential impact on health. Their use allows for the assessment of heatwave events.

#### 2.2.2. Health Data

Health data were obtained from Ourossogui Regional Hospital through field surveys aimed at digitizing medical records and collecting anonymized patient information, along with diagnoses of pathologies potentially linked to heatwaves.

Given resource limitations, the dataset covers the period from January 2017 to May 2022. The anonymized patient records include essential variables such as age, sex, date of admission, and primary diagnosis. A rigorous data cleaning process was conducted to exclude cases unrelated to climatic factors, such as road traffic accidents, fractures, scorpion stings, and snake bites.

The selection of heat-sensitive pathologies was carried out in collaboration with medical professionals to ensure relevance. The final dataset includes conditions potentially exacerbated by extreme temperatures, such as cardiovascular diseases, respiratory disorders, metabolic and renal impairments, infectious and febrile illnesses, heat-related syndromes, and other severe conditions.

### 2.3. Detection and Characterization of Heatwaves

#### 2.3.1. Identification of Heatwaves

In this study, we used TMAX and the daily maximum heat index (HI) to detect heatwaves.

TMAX is a key indicator due to its direct link with heat stress and health impacts [46]. It reflects heat intensity and helps identify periods of extreme heat [47,48].

HI is used to estimate the perceived temperature by incorporating relative humidity. The equation developed in the studies of Rothfusz [49], based on the work of Steadman [32], is commonly employed for this calculation. In dry climates, low humidity enhances evaporation and reduces thermal discomfort, whereas in humid climates, high humidity hinders sweat evaporation, exacerbating heat stress [50,51].

The Rothfusz equation used for HI is:
(1)
$$\begin{array}{l} HI=-42.379+2.04901523\times T+10.14333127\times R-0.22475541\times T\times R\\ \;\;\;\;\;\;\;\;\;\;\;\;\;\;\;\;\;\;\;\;\;\;\;\;\;-0.00683783\times T^{2}-0.05481717\times R^{2}+0.00122874\times T^{2}\times R\\ \;\;\;\;\;\;\;\;\;\;\;\;\;\;\;\;\;\;\;\;\;\;\;\;\;+0.00085282\times T\times R^{2}-0.00000199\times T^{2}\times R^{2} \end{array}$$


When R < 13% and 27 °C (80 °F) ≤ T ≤ 49 °C (120 °F), an adjustment is applied to correct the drying effect of the air:
(2)
$$ADJUSTMENT=(\frac{13-R}{4})\times \sqrt{\frac{17-\left|T-95\right|}{17}}$$

where 13% is the threshold at which air becomes extremely dry [52], and 95 °F (35 °C) is where human thermoregulation begins to be significantly affected [32,49].

When R > 85% and 27 °C ≤ T ≤ 30 °C, another correction is applied:
(3)
$$ADJUSTMENT=(\frac{R-85}{10})\times (\frac{87-T}{5})$$

where 85% is the threshold at which air becomes saturated with humidity, reducing sweat evaporation, and 87 °F (30 °C) is a reference point for perceived heat [53].

When T < 27 °C (80 °F), a simplified equation is used:
(4)
$$HI=0.5\times \left\{T+61+[(T-68)\times 1.2]+(R\times 0.094\left.)\right\}\right.$$


If HI > 80 °F (27 °C), the full Rothfusz equation is applied to ensure a more precise estimation of the perceived temperature.

Integrating HI into heatwave detection enhances the accuracy of assessments, particularly in environments where humidity plays a critical role in thermal perception. In this study, 
T
 represents TMAX, and 
R
 represents RHMIN. Assuming the atmospheric moisture content remains constant, an increase in temperature generally leads to a decrease in relative humidity, which typically reaches its minimum at the peak temperature. Based on this assumption, we match the TMAX with the RHMIN to calculate the HI.

The methodological approach is based on the 90th percentile (90thP), calculated over the reference period 1991–2020 and smoothed using a 7-day moving window. A heatwave is thus defined as a period of at least three consecutive days during which the indicator exceeds this threshold.

#### 2.3.2. Heatwave Characterization Variables

After detecting heatwaves, it is essential to extract the most relevant characteristics to understand their impact on human health. These variables help identify key indicators that influence physiology and heat-related pathologies. Table 1 defines the main variables used to characterize these events:

The identification of these variables is based on their ability to capture climatic fluctuations that may impact human health. They help assess the duration, intensity, and dynamics of heatwaves and understand their pathophysiological effects.

Heatwave Presence (HW): This variable is essential for identifying periods when the temperature exceeds a critical threshold for several consecutive days. Studies have shown that prolonged heatwaves are strongly associated with increased mortality and morbidity, particularly among the elderly and individuals with chronic diseases [54]. Prolonged exposure to high temperatures increases heat stress, potentially leading to cardiovascular decompensations and metabolic complications [55].

Duration (D): A key factor influencing health risks. Prolonged exposure exacerbates heat stress, increasing dehydration and exhausting thermoregulation mechanisms [56]. Studies [57] have demonstrated that heatwaves lasting more than five days significantly increase the risk of hospitalization for acute kidney failure and cardiovascular diseases, mainly due to fluid loss and vascular dysfunction.

Mean Intensity (
$I_{mean}$
): Reflects the overall thermal burden experienced by the body during a heatwave [58]. A high 
$I_{mean}$
 can lead to severe hyperthermia, heatstroke, and heat exhaustion [59].

Peak Temperature (
$I_{max}$
): Represents a critical risk factor [60]. A sudden rise in temperature places significant pressure on the cardiovascular system, increasing the risk of strokes and myocardial infarction [61]. A maximum temperature exceeding the seasonal threshold by 5 °C is correlated with an increase in hospital admissions for acute coronary syndromes [62].

Daily Variability of Excess Temperatures (
$I_{var}$
): Rapid fluctuations in temperature during an extreme event can worsen physiological stress [63]. Large temperature variations are associated with metabolic dysfunction and an increased risk of electrolyte imbalances [64]. The human body struggles to adapt to sudden temperature changes, which may worsen respiratory and neurological conditions [65].

Cumulative Intensity (
$I_{cum}$
): Measures heat accumulation throughout the event. The higher this value, the greater the health impact, as the body’s compensatory mechanisms become progressively exhausted [66]. Research [67] has shown a correlation between cumulative intensity and increased hospital admissions for chronic respiratory diseases, mainly due to prolonged airway inflammation.

Rate of Onset (RO): Quantifies the speed at which the temperature reaches its peak. A sudden increase in heat leaves little time for physiological adaptation, increasing the risk of acute heat stress and respiratory distress [68]. Interviews with healthcare personnel confirmed that this phenomenon is particularly dangerous for individuals with pre-existing conditions.

Rate of Decline (RD): A rapid drop in temperature after the peak can also be harmful. A sudden decrease disrupts physiological regulation mechanisms, increasing the risk of neurovascular disorders [69]. During a heatwave, the body gradually develops thermal tolerance and adjusts its physiological responses to cope with extreme conditions. However, a sudden drop in temperature can cause abrupt imbalances [70].

The health impacts of heatwaves do not always occur immediately but can extend several days after the event [71]. To better understand these delayed effects, we incorporated lag variables, which track hospitalization trends from 1 to 7 days after a heatwave (
${Lag}_{k}(x)$
, where x represents the heatwave variable and k is the lag in days). Studies [72] indicate that health consequences peak between Day +3 and Day +5 after a heatwave, as pathophysiological complications take time to manifest. Other research [24,73] confirms that heat stress triggers delayed health effects, particularly affecting cardiovascular and respiratory diseases.

### 2.4. Selection of Explicit Variables

Several studies have demonstrated that age and gender are key vulnerability factors in the face of heatwaves [74,75,76,77]. Our field surveys and interviews with healthcare personnel confirmed these observations:Differences in Hospital Attendance: Although men generally visit healthcare facilities more frequently, heatwaves lead to an increase in female consultations, suggesting higher vulnerability among women;At-Risk Groups: Infants and dependent elderly individuals, despite benefiting from family protection, remain the most vulnerable to the effects of heatwaves.

To integrate these findings into our analysis, we added additional variables, including the following:Number of female patients consulted per day;Number of infants (0–4 years) consulted per day;Number of fragile seniors (65–79 years) consulted per day;Number of very elderly and dependent seniors (80+ years) consulted per day.

These variables enhance predictions by accounting for differential vulnerabilities among exposed populations.

A total of 68 climate and vulnerability-related variables were identified. However, not all of them are necessarily relevant for predictive model construction. To reduce overfitting and improve model robustness, we applied the Least Absolute Shrinkage and Selection Operator (LASSO) method.

Developed by Robert Tibshirani [78], this variable selection technique applies an L1 penalty to regression coefficients, automatically eliminating non-significant variables and reducing model complexity.

The LASSO is defined as follows:
(5)
$$\hat{\beta }={\arg min}_{\beta }(\sum_{i=1}^{n}{\left(y_{i}-X_{i}\beta \right)}^{2}+\lambda \sum_{j=1}^{p}\left|\beta_{j}\right|)$$

where 
$\;y_{i}$
 is the dependent variable (observed values), 
$X_{i}$
 is the matrix of explanatory variables, 
$\beta_{j}$
 represents the coefficients of the explanatory variables, λ is the regularization parameter, which controls the strength of the penalty, n is the number of observations, and p is the total number of explanatory variables.

For interpretation, if λ = 0, the LASSO estimator is equivalent to a classical linear regression without regularization. If λ is large, many coefficients 
$\beta_{j}$
 will be reduced to zero, leading to an automatic selection of the most relevant variables.

To ensure that the scale of the variables does not influence the selection, a standardization of the explanatory variables was applied. Then, cross-validation was performed to optimize the parameter λ by selecting the value that minimizes the prediction error. This process balances the minimization of the mean squared error and the penalization of coefficients, thereby ensuring a more reliable model.

Once λ was optimized, only variables with nonzero coefficients were retained as the most influential in explaining hospitalizations related to heatwaves. Furthermore, the selected variable was prioritized along with its 7 lags to facilitate the interpretation of the delayed effects of heatwaves on hospital admissions.

### 2.5. Optimization and Evaluation Method of Models

After selecting the most relevant explanatory variables, the dataset was divided into two subsets:Training set (80%): Used to fit and train the models;Test set (20%): Used to evaluate model performance.

To ensure a reliable model evaluation and avoid overfitting, a 10-fold cross-validation approach was implemented. This method divides the data into 10 subsets (using 9 for training and 1 for testing, repeated 10 times), which helps reduce variance in performance estimates by testing the model on multiple sub-samples.

After this phase, model hyperparameters were optimized using Grid Search combined with cross-validation to fine-tune the key parameters and maximize predictive performance.

Model performance was evaluated using three metrics:Coefficient of Determination (R^2^): Measures the proportion of variance in the observed data explained by the model [79].Root Mean Square Error (RMSE): Indicates the dispersion of prediction errors, heavily penalizing large errors [80].Mean Absolute Error (MAE): Computes the average magnitude of absolute differences between predicted and observed values, offering a direct interpretation of average error magnitude [81].

The combined use of these metrics (Table 2) provides comprehensive insights into model performance. Specifically, R^2^ assesses the overall explanatory power and global fit of the model [82]. RMSE, by emphasizing larger errors, is especially valuable for identifying significant deviations and understanding error dispersion. MAE, less sensitive to outliers, provides a robust measure of average error magnitude [83]. This combination aligns with best practices recommended in the predictive modeling literature, enabling effective evaluation of both explanatory capability and error distribution [83,84].

Bootstrapping was used to estimate the variability and robustness of the model performance metrics while avoiding reliance on single-point estimates. This method is based on resampling with replacement, allowing the generation of multiple independent subsamples from the original dataset. This approach mitigates dependency on a single training sample and reduces biases associated with data variability, ensuring a more reliable estimation of model performance. We applied 1000 iterations to ensure stable convergence of the estimates, following existing study recommendations [85,86,87]. At each iteration, a new sample was drawn with replacement, and performance metrics were recalculated. This process generated an empirical distribution, enabling the estimation of 95% Confidence Intervals (CIs) for each metric.

The primary advantage of bootstrapping is its ability to assess model robustness in response to data variations. A wide confidence interval indicates increased sensitivity to sample fluctuations, suggesting a more unstable model or a tendency toward overfitting [88]. Conversely, a narrow confidence interval reflects better generalization of the model to new data.

Finally, a variable importance analysis is conducted to identify the key factors influencing hospitalizations. This analysis helps determine the variables that have the greatest impact on hospital admissions.

We selected two machine learning models, Random Forest (RF) and Extreme Gradient Boosting (XGB), as well as a statistical model, namely Generalized Additive Models (GAMs).

The RF model relies on an ensemble of decision trees trained on random samples and subsets of variables. This approach reduces variance and mitigates the risk of overfitting while providing variable importance measures to facilitate interpretation [89].

XGB employs a gradient boosting method with regularization (L1/L2) and internal optimizations (parallelization, memory management). It is fast and often highly performant, making it a popular choice in data science competitions [90].

GAM [91] was chosen for its ability to model nonlinear relationships by incorporating temporal smoothing effects using 50 splines, as in the work of Wood [92]. A Poisson distribution with a logarithmic link function was used, making it suitable for counting data such as hospitalizations [93,94].

## 3. Results

The TMAX index was used to identify heatwaves without accounting for humidity effects. This approach allows for a clearer visualization of the variability of extreme events. As shown in Figure 2a, 30 distinct heatwave events were recorded between 2017 and 2022, with durations ranging from 3 to 14 days. The longest event occurred from 2 to 15 March 2018, lasting 14 days, with TMAX values ranging from 42.6 °C to 45.5 °C and relatively low humidity (20–25%). These dry and hot conditions suggest potential interactions with dust episodes, as highlighted by Diouf et al. [41], which may aggravate respiratory pathologies.

The most intense TMAX-based heatwave was observed in March 2020, lasting five days and reaching a peak intensity of 5.56 °C, with maximum temperatures up to 48 °C. Most events cluster between March and June, as well as in October–November, coinciding with seasonal transitions. These periods of extreme heat under dry conditions are particularly dangerous for children and outdoor workers due to increased risks of dehydration, heatstroke, and cardiovascular stress.

In contrast, heatwaves identified using the heat index (HI), which integrates both temperature and humidity, show a different temporal and intensity distribution (Figure 2b). Only 11 events met the detection criteria, with a slight but notable increase in their frequency and intensity observed since 2020, suggesting a gradual intensification of heatwave occurrences in recent years. HI-based heatwaves lasted between 3 and 9 days. The most intense one occurred from 26 to 31 January 2021, with an intensity of 4 °C, TMAX between 37 and 41 °C, and relative humidity between 16% and 23%.

Interestingly, HI-based events mostly occur in months with residual humidity, January, February, March, August, September, November, and December, suggesting that humidity retained after the rainy season may significantly increase thermal discomfort. This result is consistent with the Sahelian context of Matam, where overall humidity is lower than in western Senegal, thus reducing the frequency of HI-detected heatwaves.

Additionally, analyzing the most extreme heatwaves using TMAX and HI allows for identifying the longest and most intense episodes, providing key insights for assessing associated health risks (see Figure 3).

Panel (a) shows the longest heatwave based on TMAX, lasting 14 days from 2 to 15 March 2018, with TMAX values ranging between 42.6 °C and 45.5 °C under low humidity conditions (10–15%) and 
$I_{max}$
 of 1.54 °C. Panel (b) illustrates the most intense TMAX-based event, which occurred from 4 to 8 March 2020, with a peak temperature of 48 °C and 
$I_{max}$
 of 5.56 °C. This event was also characterized by a sharp rate of decline (RD) of 6.1 °C/day.

Panels (c) and (d) present the longest and most intense heatwaves based on the HI, respectively. The longest HI-based event spanned 9 days (2–10 March 2018), with 
$I_{max}$
= 1.2 °C and a 
$I_{cum}$
 of 4.6 °C. The most intense HI-based heatwave occurred from 26 to 31 December 2020, with 
$I_{max}$
 = 4 °C and a notable temperature drop marked by RD = 2.5 °C/day.

The impact of heatwaves on health can be assessed through hospitalizations recorded during the study period. A descriptive analysis of the collected data by age group and gender helps identify the most affected populations.

Figure 4a shows that hospitalization rates per 1000 inhabitants increase markedly with age. Individuals aged 80 years and older had the highest standardized hospitalization rate (7.65‰), followed by those aged 65–79 years (6.17‰) and 45–64 years (5.21‰). This trend highlights the elevated vulnerability of elderly populations to adverse health outcomes, likely due to a higher prevalence of chronic conditions, reduced thermoregulatory capacity, and increased frailty.

Conversely, younger age groups exhibited lower hospitalization rates: 4.25‰ for the 18–44 years group, 2.48‰ for children aged 0–4 years, and 1.66‰ for the 5–17 years group. Despite their physiological sensitivity, the lower hospitalization rate among young children may be explained by greater family care, underreporting, or limited access to hospital facilities. The pattern confirms that using standardized rates provides a clearer view of relative vulnerability across age groups, which is obscured when using raw hospitalization counts alone.

Regarding gender differences (Figure 4b), men showed a higher standardized hospitalization rate (4.71‰) compared to women (4.24‰). This disparity can be attributed to men’s greater representation in occupations that involve high exposure to heat (e.g., agriculture, transport, informal labor), combined with sociocultural factors that may delay healthcare-seeking behavior. Although women are biologically more vulnerable to heat-related conditions, their earlier and more frequent engagement with health services may partly explain their lower hospitalization rate.

The application of LASSO variable selection revealed that 
$I_{cum}$
 and its lagged variables are the most predictive. Therefore, these variables were selected as the main climatic features (TMAX-based events), complementing the already-defined demographic vulnerability variables.

The model evaluation presented in Table 3 highlights differences in predictive performance. RF emerges as the most effective model, with an R^2^ ranging from 0.51 to 0.72, RMSE between 0.91 and 1.38, and MAE between 0.74 and 0.89. XGB follows closely, with an R^2^ between 0.46 and 0.72, RMSE between 0.91 and 1.46, and MAE between 0.74 and 0.90, indicating a similar level of accuracy but with slightly higher variability. In contrast, GAM demonstrates lower performance, with an R^2^ ranging from 0.33 to 0.48, RMSE between 1.22 and 1.50, and MAE between 0.89 and 0.98, confirming its limited ability to capture the complex interactions between heatwaves and hospital admissions.

The 95% CI obtained through bootstrapping reveals different levels of variability among the models. RF has a CI width of 0.21, indicating good stability, although some variations exist depending on the data sampling. XGB has a slightly wider CI (0.26 on R^2^), suggesting a higher sensitivity to data variations. On the other hand, GAM has a narrower CI (0.15 on R^2^), but this does not indicate greater robustness. Instead, it reflects rigidity in modeling, limiting its adaptability to nonlinear dynamics.

These results confirm that machine learning models outperform GAM. RF stands out for its robustness and balance between accuracy and stability, making it the best choice for predicting heatwave-related hospitalizations. XGB remains a competitive alternative, with performance close to RF, though slightly more sensitive to data variations. GAM, while useful for interpretability, struggles to capture the complexity of climate-health interactions, reducing its predictive relevance.

We observed that only 11 heatwave events were detected using HI, and the application of LASSO identified a limited number of explanatory variables. Furthermore, when applying the models, all R^2^ values were negative, and the RMSE and MAE values were high, indicating that the model fails to adequately explain the relationship between hospitalizations and HI-based heatwaves.

This limitation justifies the use of TMAX, which provides a relatively better explanatory capacity. This difference could be attributed to the short study period, which may have affected the robustness of the HI-based analysis.

Figure 5a illustrates the coefficients derived from the GAM model. The immediate effect of cumulative heatwave intensity (
$I_{cum}$
) appears positive (coefficient = 0.0203), suggesting a potential increase in hospitalizations. However, this result is not statistically significant (*p* = 0.1117), preventing any conclusive interpretation regarding the immediate impact of heatwaves on hospital admissions.

Regarding delayed effects, only Lag5 shows a statistically significant association (*p* = 0.0361), with a negative coefficient (−0.0334), indicating a decrease in hospitalizations five days after a heatwave event, an unexpected outcome. Other lags (Lag1 to Lag4, Lag6, and Lag7) are not statistically significant (*p* > 0.05), offering limited evidence of delayed effects.

This isolated finding for Lag5 may be due to methodological limitations. In particular, GAM models tend to underperform when the time series is short or when the number of predictors is relatively large [95], making them less suited to capturing complex temporal dynamics in hospitalization patterns.

Figure 5b presents the variable importance derived from the RF model, measured by the percentage increase in mean squared error (%IncMSE) following permutation. A higher value indicates a greater contribution of the variable to model performance.

The results highlight that delayed effects are more influential than immediate effects, with a peak on the third day (Lag3 = 183.52), a slight decrease on the fourth day (Lag4 = 157.73), and a subsequent rebound on the fifth day (Lag5 = 172.32).

This temporal pattern may be explained by behavioral and physiological delays. Interviews with healthcare personnel suggest that during and immediately after a heatwave, vulnerable individuals tend to avoid hospitals due to extreme temperatures, either attempting to self-manage their symptoms or being physically unable to travel. Consequently, those requiring care may delay their hospital visit until symptoms worsen, often around the third day.

The decline observed on day four may reflect the fact that some patients were already hospitalized the previous day. The secondary peak on day five could be attributed to the delayed onset of heat-related complications, particularly cardiovascular conditions, which typically take 72 h to manifest severe symptoms.

After day five, the influence of heatwaves on hospitalizations diminishes, with Lag6 (93.99) and Lag7 (59.55) showing relatively lower importance. The immediate and early lags (
$I_{cum}$
 = 59.80; Lag1 = 69.31; Lag2 = 75.77) contribute less to model accuracy, reinforcing the conclusion that delayed effects are more pronounced than immediate ones.

These findings underscore the need for post-heatwave monitoring of hospital admissions over at least five days and suggest that health preparedness strategies should incorporate delayed health risks, not only immediate responses.

Figure 5c displays the variable importance from the XGB model using the Gain metric, which quantifies the relative contribution of each variable to model accuracy. Results indicate that Lag3 (Gain = 0.021) and Lag5 (Gain = 0.022) are the most influential predictors, further supporting the presence of delayed effects three to five days following a heatwave.

This delay is likely linked to pathophysiological mechanisms, in which heat stress triggers complications that take several days to reach clinical severity. In contrast, the immediate effect (lag0) and early lags (Lag1 and Lag2) display lower Gain values, suggesting a limited increase in hospital admissions on the day of or immediately following a heatwave. Beyond Lag5, the importance of predictors declines (Lag6 and Lag7), indicating a fading impact of heat exposure over time.

To refine the interpretation of machine learning predictions, we applied the SHAP (Shapley additive explanation) method developed by Lundberg and Lee [95], which quantifies the contribution of each feature to individual predictions.

Figure 6 presents the SHAP values of the main predictors for the RF (a) and XGB (b) models. In both cases, lagged variables such as Lag3 and Lag5 emerged as the most influential, while immediate effects (Lag0, Lag1) appeared less determinant. These results underscore the importance of delayed effects in explaining heatwave-related hospitalizations in Matam. However, it is worth noting that the absolute SHAP values remain relatively low, suggesting limited individual predictor impact.

The overall impact of the variables is more pronounced in the RF model, reflecting a higher sensitivity to predictor structure. The SHAP analysis thus confirms the relevance of the selected variables and enhances the individual-level interpretability of the models.

Figure 7 highlights the joint evolution of TMAX and hospital admissions over 7 days before and 14 days after a heatwave event (from 15 June to 21 June 2019), during which 
$I_{cum}$
 was equal to 14 °C. During this period, several temperature peaks reaching 46 to 48 °C were recorded, significantly increasing thermal stress on the population. Examination of the curves reveals a correlation between high temperatures and hospital admissions. Indeed, the blue curve representing hospital admissions shows an increase a few days following periods of extreme heat. This trend suggests a delayed effect of heatwaves on hospitalizations, potentially due to the gradual worsening of underlying conditions and the slow physiological response of patients exposed to heat. This delayed effect is consistent with observations by Kenny et al. [96], who reported that heat stress could trigger medical complications within a period varying from one to five days following exposure.

The trend in hospital admissions also indicates that certain peaks occur after the hottest days, which the behavior of exposed populations could explain. Vulnerable individuals might delay seeking medical care until their symptoms become severe, possibly due to long distances to healthcare facilities or varying perceptions of risk among different social groups. This observation is particularly concerning for outdoor workers and elderly populations, whose ability to adapt to heat is limited.

Finally, 6 to 7 days after the heatwave, a decrease in hospital admissions is observed in parallel with the gradual decline in temperatures. This suggests a return to more physiologically tolerable conditions, leading to stabilization in hospital admissions. These results confirm the importance of establishing surveillance and early warning systems to anticipate hospitalization peaks associated with heatwaves and implement preventive measures tailored to at-risk populations.

These observations concretely illustrate the impact of heatwaves on hospital morbidity, reinforcing the need to integrate climate parameters into health management strategies.

## 4. Discussion

This study highlighted a correlation between rising extreme temperatures and increased hospital admissions. The findings demonstrated a clear relationship between extreme temperatures and hospital attendance rates. Specifically, a delayed effect was observed, with heatwave-related hospital admissions peaking several days after the heat event itself, possibly due to symptom worsening or delayed care-seeking behavior influenced by distance to healthcare facilities and varied risk perception among social groups. This delayed response is especially concerning for outdoor workers and elderly populations, whose adaptive capacities to heat stress are limited.

Machine learning models outperformed traditional approaches, with RF achieving the highest accuracy (R^2^ = [0.51; 0.72]) in capturing nonlinear relationships and delayed effects. XGB also demonstrated strong predictive capabilities. Model robustness was assessed using bootstrapping (1000 iterations) to estimate 95% CI, confirming RF’s stability (CI = 0.21) compared to XGB (CI = 0.26). While GAM exhibited a narrower CI (0.15), it reflected structural rigidity rather than better robustness, limiting its ability to model complex interactions.

These findings align with previous research: Kenny et al. [96] highlighted delayed medical complications from heat stress, while Amegah et al. [5] and Deng et al. [60] confirmed increased hospitalizations due to heat-related conditions such as heatstroke, dehydration, and cardiovascular complications.

Distinct regional characteristics emerged, particularly comparing Matam and Dakar. Unlike Dakar, where humidity exacerbates the sensation of heat, Matam experiences dry heat, potentially accelerating dehydration and worsening existing health conditions. Furthermore, limited healthcare infrastructure access in Matam amplifies vulnerability and complicates prompt healthcare interventions, exacerbating the delayed health impacts observed.

Physiologically, cardiovascular and respiratory conditions triggered by heat stress may require latency periods before hospital admission becomes necessary. Behavioral factors, including reluctance to seek care during extreme heat to avoid additional exposure, may also delay medical interventions. Limited healthcare infrastructure and the underestimation of heat-related risks by local populations contribute further to these delays.

Methodologically, integrating demographic vulnerability factors (number of women, infants, and dependent elderly individuals) and temporal lag variables enhanced model accuracy, emphasizing the critical need to consider demographic factors and temporal dynamics in heat-health studies, elements often overlooked in previous research.

However, the study acknowledges limitations: potential inaccuracies or diagnostic errors due to handwritten hospital records, and the exclusion of environmental factors such as air pollution or Saharan dust, which could amplify heatwave health impacts.

These findings underline the importance of establishing effective surveillance and early warning systems to anticipate hospitalization peaks linked to heatwaves. Integrating climate forecasts into health response plans would enable better management of heat-related health risks. Preventive measures should specifically target vulnerable populations, elderly, children, and outdoor workers, by ensuring adequate hydration, improving infrastructure ventilation and air conditioning, and training healthcare providers for emergency situations.

Hospitals must reinforce emergency preparedness by stocking essential supplies (water, rehydration solutions, electrolytes), improving facility ventilation and air conditioning, and training healthcare staff. Targeted interventions for vulnerable groups, particularly the elderly, children, and outdoor workers, should be prioritized.

Future research directions include extending studies to other health facilities and regions to allow comparative analyses. Advanced deep learning techniques could further enhance prediction accuracy by integrating complex datasets and modeling nonlinear interactions more effectively.

## 5. Conclusions

This study underscores the significant relationship between extreme heat and hospital admissions in Matam, Senegal, using a combination of meteorological data and machine learning approaches. The observed time lag between heatwaves and peaks in hospitalizations highlights the need for anticipatory health interventions and context-specific preparedness strategies. Among the models tested, RF demonstrated the highest predictive power, confirming the potential of machine learning for modeling complex climate–health interactions, especially when traditional models like GAM underperform in data-scarce settings.

These findings support the integration of climate information into health surveillance systems to improve early warning capabilities and decision-making processes. They also point to the need for context-adapted interventions, particularly in regions with limited healthcare infrastructure. Future research should aim to incorporate additional environmental and socioeconomic determinants, expand spatial coverage across other vulnerable regions, and leverage advanced learning methods such as deep learning to enhance prediction accuracy and generalizability further.

## Figures and Tables

**Figure 1 ijerph-22-01349-f001:**
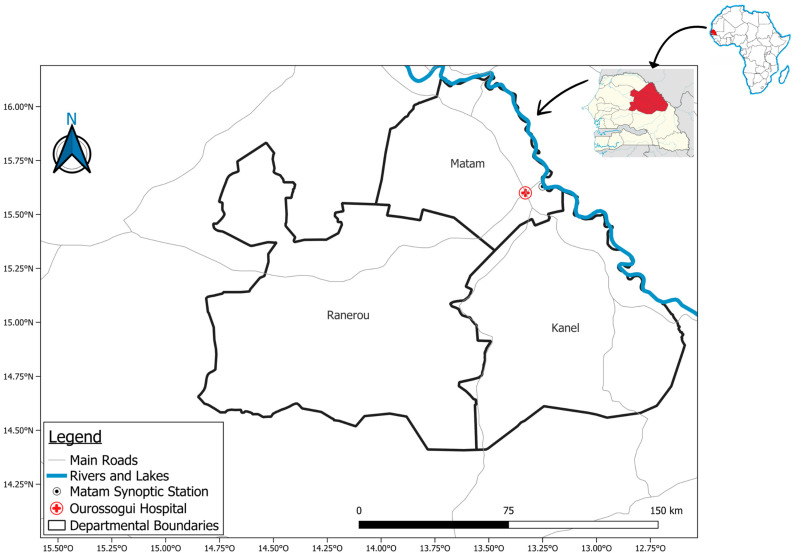
Geographical location of Matam region and key infrastructure.

**Figure 2 ijerph-22-01349-f002:**
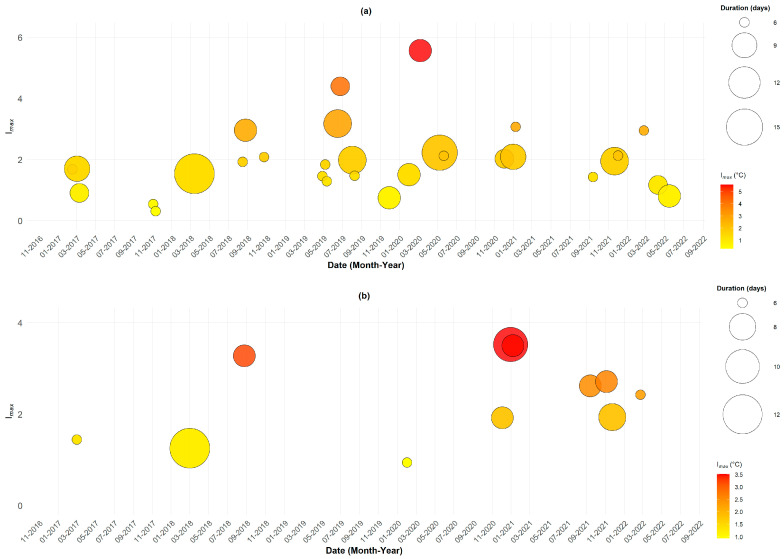
Heatwaves detected with TMAX and HI (2017–2022). Each point represents a heatwave event, with the x-axis showing the peak date and the y-axis the maximum intensity (*I_max_*). Circle size reflects event duration (in days), while the color scale indicates intensity, from yellow (low) to red (high). (**a**) TMAX-based events. (**b**) HI-based events.

**Figure 3 ijerph-22-01349-f003:**
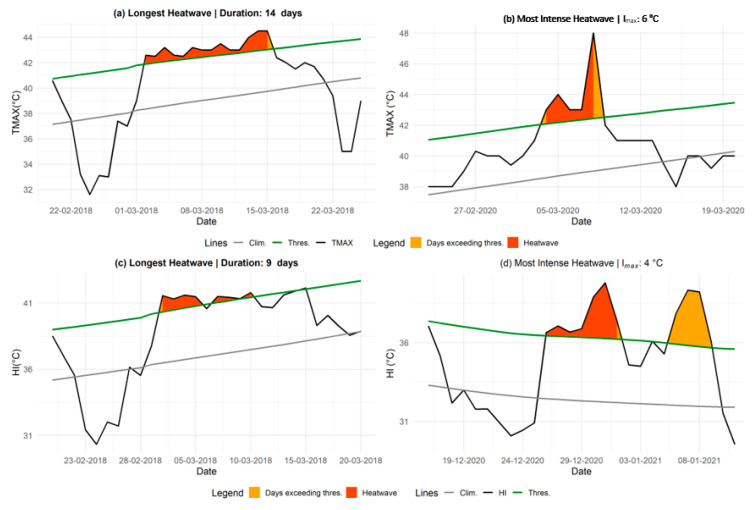
Longest and most intense heatwaves based on TMAX and HI. Panels (**a**,**c**), respectively, depict the longest heatwaves, while panels (**b**,**d**) illustrate the most intense episodes observed in Matam based on TMAX and HI indices. The orange-shaded areas indicate periods when the index (TMAX or HI) exceeded the threshold, turning red if the duration reached at least three consecutive days. The black line shows the daily index values, the green line denotes the heat-wave threshold, and the grey line represents the daily climatology.

**Figure 4 ijerph-22-01349-f004:**
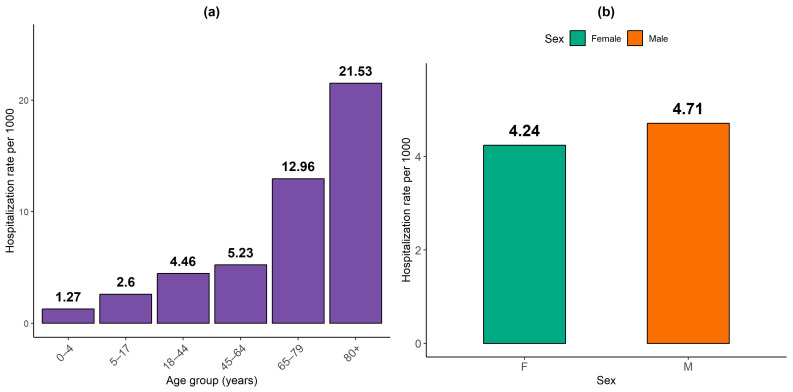
Distribution of standardized hospitalization rates by age group and sex (January 2017–May 2022) at Ourossogui Hospital. (**a**) Age-standardized hospitalization rates. (**b**) Sex-standardized hospitalization rates.

**Figure 5 ijerph-22-01349-f005:**
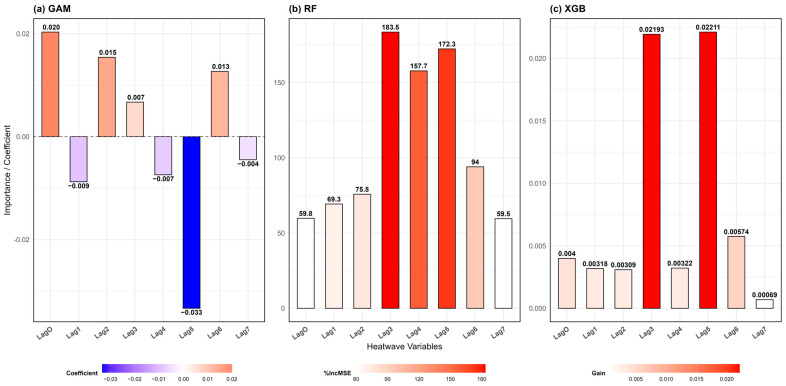
Variable importance of heatwave-related predictors in the (**a**) GAM, (**b**) RF, and (**c**) XGB models. In GAM, importance is expressed as regression coefficients; in RF, as percentage increase in %IncMSE; and in XGB, as Gain values. Warmer colors (red) indicate stronger contributions, while lighter tones (white to blue) represent lower influence.

**Figure 6 ijerph-22-01349-f006:**
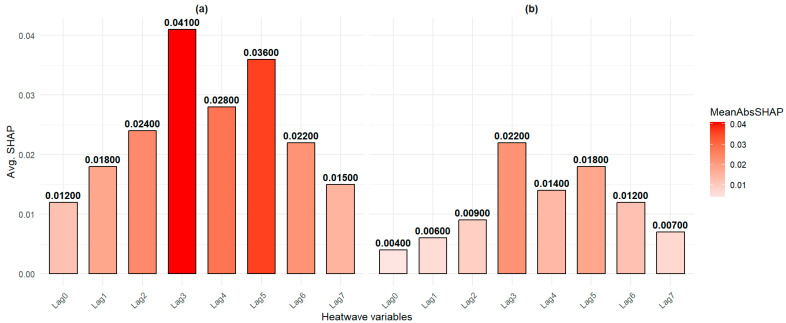
SHAP values of heatwave-related predictors in (**a**) RF and (**b**) XGB. Bars represent the average contribution of each variable to prediction accuracy across the test set. Warmer colors (red) denote higher importance.

**Figure 7 ijerph-22-01349-f007:**
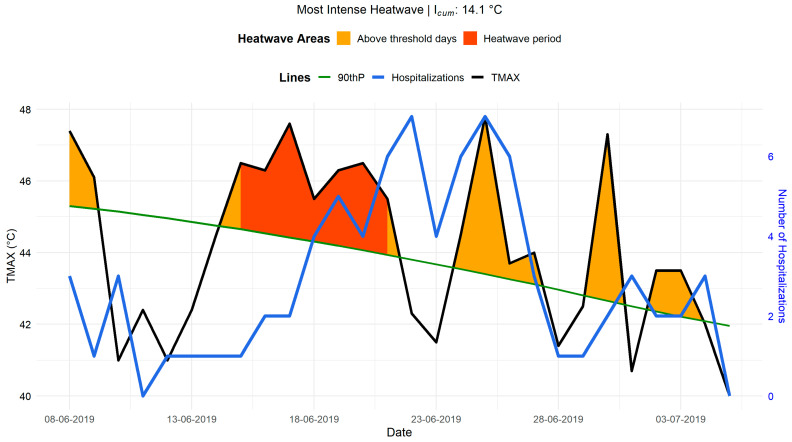
Relationship between heatwaves (15–21 June 2019) and hospital admissions. TMAX is shown in black, and hospital admissions in blue. Orange shading indicates days exceeding the temperature threshold (90th percentile, green line), and red shading marks the heatwave period.

**Table 1 ijerph-22-01349-t001:** Variables characterizing heatwaves.

Variable	Formula	Unit	Description
HW	HW = 1 if Index > 90thP for at least 3 days, otherwise 0	-	Presence of a heatwave. Indicates whether the thermal index exceeds the threshold for at least three consecutive days.
D	D=∑hw	days	Total duration of the heatwave in consecutive days.
$I_{mean}$	$I_{mean}=\frac{1}{D}\sum Index-90eP$	°C	Average excess temperature index above the threshold during the event.
$I_{max}$	$I_{max}=max(Index-90eP)$	°C	Maximum excess value reached by the temperature index during the heatwave.
$I_{var}$	$I_{var}=Var(Index-90eP)\;$	°C^2^	Daily variability of excess temperatures.
$I_{cum}$	$I_{cum}=\sum \left(Index-90eP\right)$	°C	Total heat accumulation during the event.
RO	$RO=\frac{I_{max}}{\sum (hwbefore\;the\;peak)}$	°C/day	Rate of temperature increase until the heatwave peak.
RD	$RD=\frac{I_{max}}{\sum \left(hw\;after\;the\;peak\right)}$	°C/day	Rate of temperature decrease after the heatwave peak.

**Table 2 ijerph-22-01349-t002:** Error metrics for model evaluation.

Error Metrics	Formula	Interpretation
R^2^	$1-\frac{\sum_{i=1}^{n}{\left(Y_{i}-{\hat{Y}}_{i}\right)}^{2}}{\sum_{i=1}^{n}{\left(Y_{i}-\bar{Y}\right)}^{2}}$	Higher values (closer to 1) indicate better model fit.
RMSE	$\sqrt{\frac{1}{n}\sum_{i=1}^{n}{\left(Y_{i}-{\hat{Y}}_{i}\right)}^{2}}$	Lower = better performance
MAE	$\frac{1}{n}\sum_{i=1}^{n}\left|Y_{i}-{\hat{Y}}_{i}\right|$	

where 
n
 is the total number of data points; 
${\hat{Y}}_{i}$
 is the predicted value at point *i*; 
$Y_{i}$
 is the observed value at point *i*; 
$\bar{Y}$
 is the mean of the observed values.

**Table 3 ijerph-22-01349-t003:** Model performance metrics and parameters.

Model	R^2^	RMSE	MAE	Parameters
GAM	[0.33; 0.48]	[1.22; 1.5]	[0.89; 0.98]	Family: Poisson
RF	[0.51; 0.72]	[0.91; 1.38]	[0.74; 0.89]	Trees: 5000; Optimal number of variables (mtry): 3
XGB	[0.46; 0.72]	[0.91; 1.46]	[0.74; 0.9]	Trees: 100; Max Depth: 3; Learning Rate: 0.05; Gamma: 0; Colsample bytree: 0.8; Subsample: 0.8; Min Child Weight: 1

## Data Availability

This study’s climatic and health data are available from the Senegal Meteorological Service and the Ourossogui Regional Hospital. The climatic dataset is publicly accessible, while the health data have been aggregated and anonymized for research purposes. These datasets can be shared upon request.

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
