# Peer review of "Machine Learning-Based Prediction of Heatwave-Related Hospitalizations: A Case Study in Matam, Senegal"

_ijerph, 2025, doi:10.3390/ijerph22091349_

Round 1

Reviewer 1 Report

Comments and Suggestions for Authors

This study analyzes the impact of heatwaves using climatic indices, such as daily maximum temperatures (TMAX) and maximum heat index (HI) on hospital admissions in the Matam region, Senegal. Three predictive models, including Random Forest (RF), Extreme Gradient Boosting (XGB), and traditional Generalized Additive Models (GAM), were developed and compared to evaluate their effectiveness. Key findings include a significant delayed increase in hospitalizations occurring approximately three to five days after heatwave events. The work addresses an important environmental issue and is within the scope of a journal. The manuscript can be improved.

Abstract:

Abstract should be shortened. It is too long.

Introduction:

The introduction includes relevant information about the topic.

L 174-175: Figure 1 needs to be improved. For someone not familiar with Senegal this map is not clear.

Results:

Language can be improved:

L 420: "a notable intensification in recent years" could be explained

Please be consistent with abbreviations

L522: I_cum

L586: Icum

L 451: Icum

Discussion and Conclusion:

L 643-650 and L696-702 are very similar.

The conclusions are consistent with the results.

References:

References need to be verified as they seem to be listed alphabetically and the number often does not match, and several are missing. The reference section needs detailed verification and update.

For example:

Do not match:

L 659: Deng et al. [58]. is cited, but in the Reference List, Deng et al is [19]

Missing:

L 316: Robert Tibshirani (1996)

L 396: (Wood, 2004)

L 658: Kenny et al.

Regards,

Comments on the Quality of English Language

There is some repetitiveness in the text that I suggest to improve.

Author Response

Comment 1: The abstract should be shortened; it is too long.

Response: Thank you for highlighting this. We have revised and shortened the abstract to improve its conciseness and focus on key findings, ensuring it directly states the study’s objectives, methods, main results, and implications in a more compact format.

Comment 2: Figure 1 needs to be improved; for someone not familiar with Senegal, this map is not clear.

Response: We appreciate the suggestion. We have redesigned Figure 1 and updated its caption and description (Lines 151–152) by adding clearer labels, a scale bar, and regional context to make the map more understandable to international readers unfamiliar with Senegal.

Comment 3: Language improvements, e.g., line 420: “a notable intensification in recent years” could be explained.

Response: Thank you for your suggestion. We have clarified the sentence in lines 394–395 to indicate that there has been a slight increase in the frequency and intensity of heatwaves since 2020.

Comment 4: Please be consistent with abbreviations (I_cum, Icum, I, etc.).

Response: We have carefully reviewed the manuscript to ensure that all abbreviations and variable names are consistent throughout the text, particularly in Lines 384–390 and Table 1. We standardized them, e.g., consistently using “I_cum” to represent cumulative hospitalizations, to avoid confusion.

Comment 5: L643-650 and L696-702 are very similar.

Response: We agree with the reviewer. The sections in Lines 643–650 and 696–702 were indeed repetitive. To address this, we retained the discussion and improved the conclusion (Lines 656–662) to enhance clarity and eliminate redundancy.

Comment 6: References need to be verified as they seem to be listed alphabetically, but the numbers often do not match, and several are missing.

Response: We appreciate this observation. We have thoroughly revised the reference list (Lines 675–901) to ensure consistency between in-text citations and the reference section. We also updated missing references (e.g., Robert Tibshirani, Wood 2004, Kenny et al.) and renumbered all references correctly.

Reviewer 2 Report

Comments and Suggestions for Authors

This study examines the effects of heatwaves on hospital admissions in the Matam region of Senegal by integrating climate modeling with advanced machine learning methods.To achieve this, the authors apply three predictive models—Random Forest (RF), Extreme Gradient Boosting (XGB), and Generalized Additive Models (GAM)—and evaluate the robustness and reliability of the results through a bootstrapping approach.

The following are some of my comments and suggestions:

  1. The following sentence is missing a concluding period:
    “These techniques facilitate the identification of critical periods and climate thresholds that may have significant health consequences”
  2. The manuscript lacks conciseness, and redundant descriptions are present in many places. For example, the abstract—especially the conclusion part—is overly long. It is recommended to briefly state the key findings.
    Additionally, a better balance between the number of figures and the information conveyed should be achieved to avoid information overload. Consider streamlining or merging some of the result figures to better highlight the correspondence between model predictions and observed hospitalization peaks, thereby strengthening the logical narrative.
    The variation in population size across age groups is an important factor influencing the number of hospitalizations in each group. Therefore, the reported hospitalization counts by age group should be standardized using the total population of the corresponding age group, rather than using absolute numbers.
    For instance, if the 18–44 age group is four times the size of the 65+ age group, but their hospitalization numbers are similar, using absolute values may mislead conclusions and obscure the higher per capita risk among the elderly.
    It is recommended to standardize hospitalization counts into relative hospitalization rates (e.g., per 1,000 people) to control for demographic structure and more accurately reflect the relative risk of each age group, rather than relying solely on absolute counts to infer vulnerability.
    4. Recommend to incorporate SHAP (SHapley Additive exPlanations) analysis in the current study to better interpret the impact of key features on the model predictions.

Author Response

Comment 1: The manuscript lacks conciseness, with redundant descriptions, especially in the abstract.
Response: Thank you for noting this. As per your suggestion, we have streamlined the text, especially in the Abstract and the Results section, removing redundant statements and focusing on the essential information to improve overall conciseness.

Comment 2: A better balance between figures and information is needed; some figures can be merged or streamlined.

Response: We appreciate this recommendation. We have reviewed all figures and reduced redundancy by combining related figures and focusing on the most informative ones (see Figures 2–7. This streamlining improves readability and strengthens the logical narrative of the results.

Comment 3: Hospitalization counts by age group should be standardized using the population of each age group.

Response: Thank you for this important suggestion. We have revised the Results section (Lines 433–456) to report standardized hospitalization rates per 1,000 people in each age group and sex, which more accurately reflects relative risk and avoids misleading conclusions based on absolute numbers alone.

Comment 4: Recommend incorporating SHAP (SHapley Additive Explanations) analysis to better interpret key feature impacts.

Response: Thank you for suggesting this valuable addition. We have integrated SHAP analysis in our study (Lines 545–557) to improve the interpretability of the machine learning models. The SHAP values highlight the importance of each predictor, confirming the delayed effect of heatwaves on hospitalizations and supporting our findings.

Comment 5: One sentence is missing a concluding period.

Response: Thank you for pointing that out. We have carefully reviewed the manuscript and corrected all minor punctuation and formatting issues (Line 205-206).
